# Brain insulin sensitivity is linked to adiposity and body fat distribution

Stephanie Kullmann[1,2,8], Vera Valenta[1,2,3,8], Robert Wagner [1,2,3], Otto Tschritter[4], Jürgen Machann [1,2,5], Hans-Ulrich Häring[1,2,3], Hubert Preissl [1,2,3,6,7], Andreas Fritsche[1,2,3] & Martin Heni [1,2,3 ✉]

Brain insulin action regulates eating behavior and energy fluxes throughout the body. However, numerous people are brain insulin resistant. How brain insulin responsiveness affects long-term weight and body fat composition in humans is still unknown. Here we show that high brain insulin sensitivity before lifestyle intervention associates with a more pronounced reduction in total and visceral fat during the program. High brain insulin sensitivity is also associated with less regain of fat mass during a nine year follow-up. Cross-sectionally, strong insulin responsiveness of the hypothalamus associates with less visceral fat, while subcutaneous fat is unrelated. Our results demonstrate that high brain insulin sensitivity is linked to weight loss during lifestyle intervention and associates with a favorable body fat distribution. Since visceral fat is strongly linked to diabetes, cardiovascular risk and cancer, these findings have implications beyond metabolic diseases and indicate the necessity of strategies to resolve brain insulin resistance.

[1] Institute for Diabetes Research and Metabolic Diseases of the Helmholtz Center Munich at the University of Tübingen, Otfried-Müller-Str. 10, 72076 Tübingen, Germany. [2] German Center for Diabetes Research (DZD e.V.), Ingolstädter Landstraße 1, 85764 Neuherberg, Germany. [3] Department of Internal Medicine, Division of Endocrinology, Diabetology and Nephrology, Eberhard Karls University Tübingen, Otfried-Müller-Str. 10, 72076 Tübingen, Germany. [4] Emergency Department, Marienhospital Stuttgart, Vinzenz von Paul Kliniken, Böheimstraße 37, 70199 Stuttgart, Germany. [5] Section of Experimental Radiology, Department of Diagnostic and Interventional Radiology, University Hospital Tübingen, Otfried-Müller-Str. 10, 72076 Tübingen, Germany. [6] Institute for Diabetes and Obesity, Helmholtz Diabetes Center at Helmholtz Zentrum München, German Research Center for Environmental Health (GmbH), Ingolstädter Landstraße 1, 85764 Neuherberg, Germany. [7] Department of Pharmacy and Biochemistry, Institute of Pharmaceutical Sciences, Eberhard Karls University Tübingen, Auf der Morgenstelle 8, 72076 Tübingen, Germany. [8] These authors contributed equally: S. Kullmann, V. Valenta. ✉email: martin.heni@med.uni-tuebingen.de

There is accumulating evidence that the human brain represents an insulin sensitive organ[1]. Initial studies in animals identified a crucial role of brain insulin in the regulation of food intake[2]. This function holds true in humans, where insulin affects important neuronal functions that underlie eating behavior[1]. Finally, insulin delivery to the human brain modulates food intake[3–5].

Recent imaging studies characterized a limited number of cortical and sub-cortical brain regions that respond to the peptide hormone, including the hypothalamus[6]. Insulin also impacts the functional inter-connection of these areas[7], which underlines its importance in the control of larger networks within the brain.

Of note, not every brain responds equally to insulin. A substantial number of people display an attenuated or absent insulin response, an observation often referred to as brain insulin resistance[6]. A number of factors that associate with brain insulin resistance have been identified so far. These range from alterations at the blood brain barrier to genetics[6]. Among them, obesity is the best studied in animals and humans[1,8]. Although, for most of these factors, including obesity, it is still unclear whether they are cause or consequence of brain insulin resistance.

Besides controlling higher brain functions, insulin also influences outflows that modulate peripheral energy metabolism[6]. Based on research in animals[9], experimental studies in humans suggested that brain insulin affects peripheral lipid metabolism in visceral adipose tissue[10] and liver[11]. More importantly, insulin delivery to the brain improves whole-body insulin sensitivity[12–14] by suppressing endogenous glucose production[14,15] and stimulating glucose uptake into peripheral tissues[14]. Research in animals and humans identified the hypothalamus as one crucial region for this process[13,14,16]. As brain insulin resistance also impairs the central nervous control over peripheral energy metabolism, it has been hypothesized that this impairment could result in altered substrate distribution with preferential energy accumulation in unfavorable fat depots[14,17].

Whether body fat accumulation has detrimental effects on cardiometabolic health is mainly determined by its location[18]. This observation has led to the concept of metabolic healthy obesity with energy storage mainly in the subcutaneous compartment versus unhealthy obesity, with fat accumulation mainly in the visceral space[19].

To test whether brain insulin sensitivity affects the long-term weight course and contributes to the development of healthy versus unhealthy body fat distribution, we analyzed two datasets with whole-body MR imaging available. The first comprises long-term follow-up data of 15 participants in whom brain insulin sensitivity was determined by magnetoencephalography before they entered a lifestyle intervention program. The second is a cross-sectional cohort of 112 participants with precise functional MR imaging of hypothalamic insulin action.

## Results

**Brain insulin sensitivity and change of body weight/body fat.** So far, brain insulin resistance was identified to be associated with less weight reduction during the first 9 months of lifestyle intervention in our TULIP study[20]. We started our current analyses by testing the impact of brain insulin sensitivity on body weight and body fat distribution in the years following the 24-month lifestyle intervention (Supplementary Table 3).

Participants with high brain insulin sensitivity before entering the lifestyle intervention program achieved a greater reduction in body weight and total adipose tissue (Fig. 1 and Supplementary Table 3). By contrast, brain insulin-resistant individuals showed a slight weight loss in the first 9 months of the program, and already regained body weight as well as total and visceral adipose

tissue during the subsequent months of lifestyle intervention (Fig. 1a and Supplementary Table 3).

Brain insulin responsiveness was associated with the reduction of total dietary energy intake during the lifestyle intervention with greater reduction in those with high brain insulin sensitivity ($p = 0.0060$, MANOVA brain insulin sensitivity × time).

In the long-term follow-up, 9 years after the lifestyle intervention, baseline brain insulin sensitivity was associated with less regain in body weight (Fig. 1a and Supplementary Table 3, $p \leq 0.05$, MANOVA). In addition, baseline brain insulin sensitivity was associated with a smaller increase in total adipose tissue and visceral fat content at long-term follow-up (Fig. 1b and Supplementary Table 3). The association between brain insulin sensitivity and change in visceral fat was independent of the change in total adipose tissue during the 24 months lifestyle intervention ($p = 0.0248$, MANOVA) and until 9 years follow-up ($p = 0.0065$, MANOVA). Brain insulin sensitivity was not significantly associated with changes in subcutaneous adipose tissue following the lifestyle intervention, neither unadjusted ($p = 0.07$, MANOVA; Fig. 1c and Supplementary Table 3) nor after adjustment for change in total adipose tissue during 24-month lifestyle intervention ($p = 0.2$, MANOVA).

**Hypothalamic insulin sensitivity and body fat distribution.** As insulin action in the hypothalamus is crucial for the brain-derived modulation of peripheral energy metabolism, we tested whether insulin responsiveness in this brain area associates with body fat distribution. After food intake, regional cerebral blood flow in the hypothalamus is physiologically reduced[21]. Accordingly, reduction of blood flow after intranasal insulin application indicates high brain insulin sensitivity[22]. Hypothalamic insulin responsiveness was positively associated with visceral adipose tissue (multiple linear regression model, $p = 0.0011$, $r^2 = 0.093$, Fig. 2b), i.e. persons with high insulin sensitivity of the hypothalamus had less visceral adipose tissue. This association remained statistically significant after adjustment for BMI (multiple linear regression model, $p = 0.0037$, $r^2 = 0.451$) as well as after adjustment for sex and age (multiple linear regression model, $p = 0.0170$, $r^2 = 0.437$). In contrast, insulin responsiveness of the hypothalamus was not associated with subcutaneous adipose tissue (multiple linear regression model, $p = 0.9$, $r^2 = 0.0002$, Fig. 2c). As a result, the ratio of visceral to subcutaneous adipose tissue was correlated to hypothalamic insulin sensitivity, with a more favorable ratio in those with a strong hypothalamus response to insulin (multiple linear regression model, $p = 0.0012$, $r^2 = 0.103$, Fig. 2d). This association remained statistically significant after adjustment for BMI (multiple linear regression model, $p = 0.0042$, $r^2 = 0.140$) as well as after adjustment for sex and age (multiple linear regression model, $p = 0.0327$, $r^2 = 0.669$).

Furthermore, hypothalamic insulin response was associated with glucose metabolism. HbA1c was lower in those with a strong insulin-induced reduction of regional blood flow (multiple linear regression model, $p = 0.045$, $r^2 = 0.067$). In line, fasting glucose was lower (multiple linear regression model, $p = 0.0039$, $r^2 = 0.119$) and insulin sensitivity was better, as assessed by HOMA-IR (multiple linear regression model, $p = 0.0423$, $r^2 = 0.061$).

## Discussion

Our current results indicate that brain insulin action contributes to long-term weight course as well as to the distribution of fat throughout the body.

Participants with high brain insulin sensitivity before lifestyle intervention lost more body weight and body fat during the 24 months of the program. In addition, brain insulin sensitivity

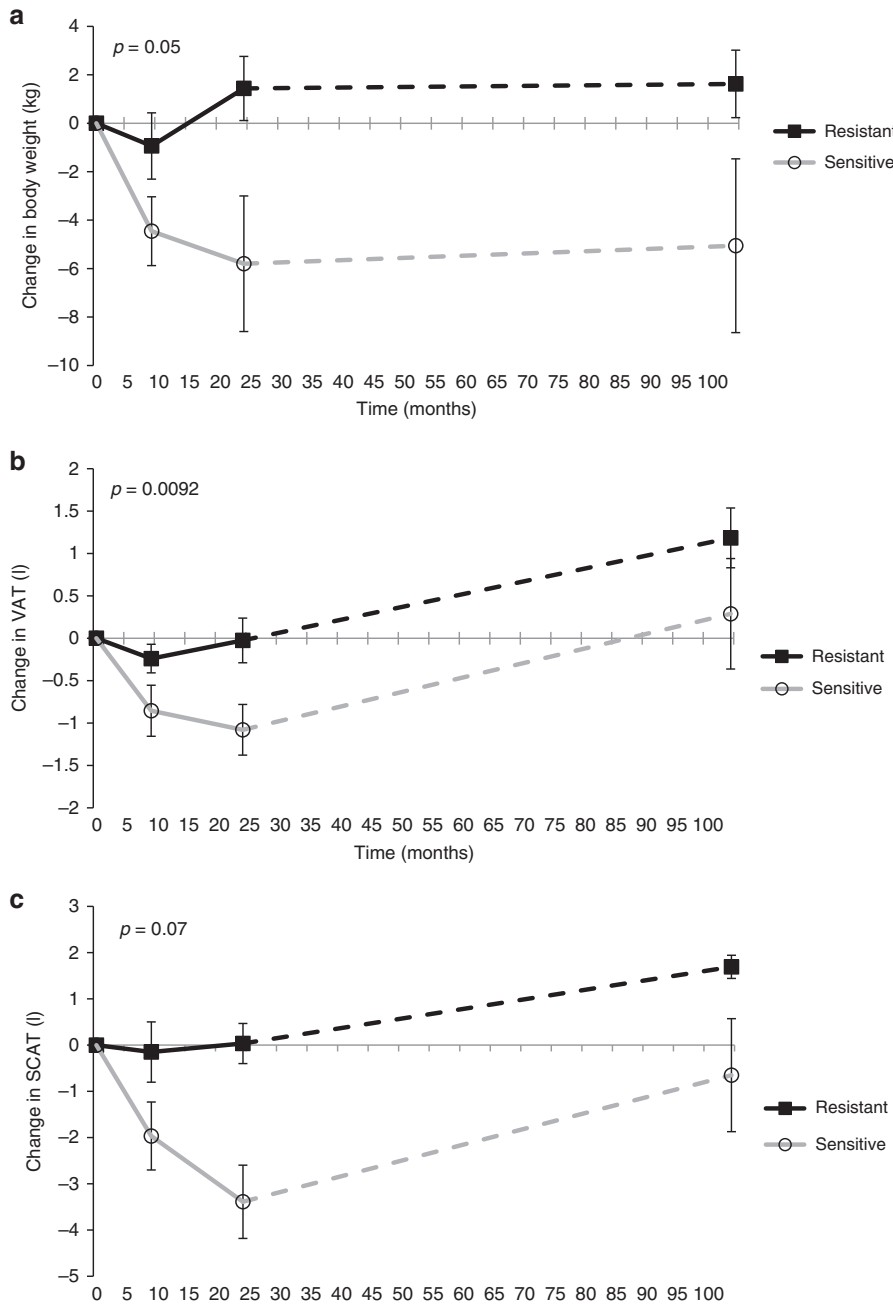

**Fig. 1 Body composition during/after 9 years of lifestyle intervention depending on brain insulin sensitivity.** (**a**) Changes in body weight; (**b**) changes in visceral adipose tissue (VAT); (**c**) changes in subcutaneous adipose tissue (SCAT). Brain insulin sensitivity was assessed as change in the theta frequency band in response to insulin infusion, corrected for saline infusion by magnetoencephalography. *p* values are from MANOVAs with brain insulin responsiveness as a continuous variable (brain insulin sensitivity × time). $N = 15$ (**a**), $N = 12$ (**b**, **c**); presented are means, error bars represent SEM. Filled boxes represent participants with brain insulin responsiveness below the median, open circles represent participants with brain responsiveness above the median. Continuous variables were used for statistical analyses and stratified variables were used solely for better illustration of the results. Source data are provided as a Source Data file.

assessed before lifestyle intervention was associated with a lower regain in body weight and body fat in the long-term follow-up. We already reported an association between high brain insulin sensitivity and immediate loss of body weight and body fat during the first 9 months of our program[20]. We now established that this relation persists throughout the entire 24 months of lifestyle intervention. Even more importantly, low brain insulin sensitivity was linked to a regain in body weight and body fat in the 9 years following the program. This association was present for the amount of visceral fat and the visceral fat content adjusted for the total amount of adipose tissue. Of note, no such association was detected for subcutaneous fat in our longitudinal data. In line with earlier data in a smaller cohort[22], our current cross-sectional analyses confirmed an association between hypothalamic insulin sensitivity and visceral fat content, but not with subcutaneous adipose tissue.

Brain insulin action impacts several brain circuitries that are crucial to eating behavior. Insulin in the human brain affects the

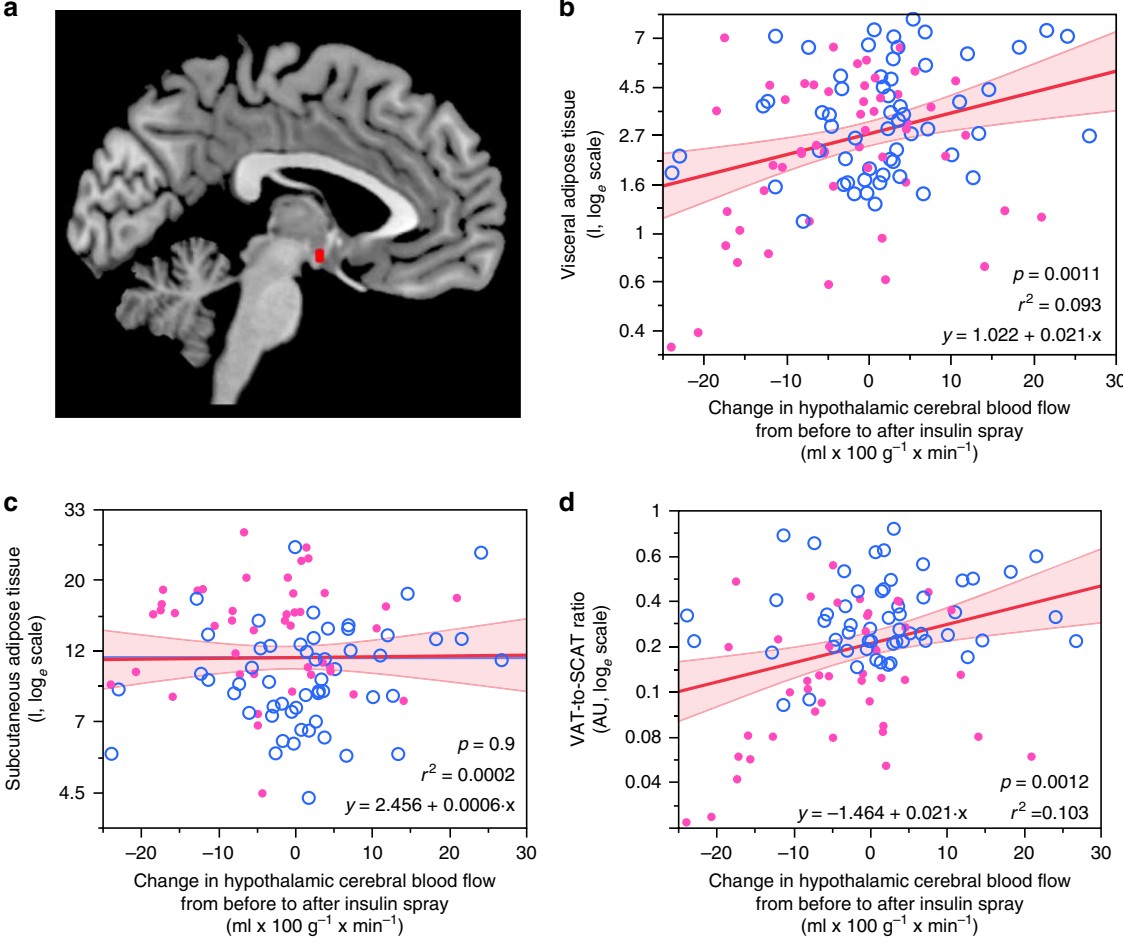

**Fig. 2 Hypothalamic insulin responsiveness associates with body fat distribution.** Region-specific change in cerebral blood flow in response to intranasal insulin administration was extracted for the hypothalamus as region of interest (**a**). Participants with a strong insulin-induced suppression in hypothalamic blood flow had significantly less visceral adipose tissue (**b**). Subcutaneous fat content was not associated with insulin sensitivity of the hypothalamus (**c**). The ratio of visceral to subcutaneous adipose tissue was favorably lower in those with strong insulin-induced hypothalamic blood flow (**d**). Pink filled circles are female participants ($N = 53$), open blue circles are males ($N = 59$). Lines represent fit lines ±CI. $p$ values are from unadjusted linear regression models. Source data are provided as a Source Data file.

response to food cues and, ultimately, food intake[5,23], mechanisms that most likely contribute to our current findings on the dietary response during lifestyle intervention and on the long-term weight regulation. However, the regulation of food choice and food intake does not appear to be the only underlying mechanisms, particularly for the association with body fat distribution. It is worth mentioning that brain insulin action modulates postprandial systemic insulin sensitivity and postprandial energy fluxes in peripheral metabolic organs via the autonomic nervous system[8,13,14,24,25]. In this context, the administration of insulin to the human brain boosts the suppression of endogenous glucose production and promotes the uptake of glucose into the peripheral organs[14]. Both these mechanisms can orchestrate postprandial energy fluxes, and help to prevent excessive energy storage in the visceral fat compartment[17] and can therefore contribute to our current findings.

This fat depot specific effects of brain insulin action could further be determined by differential autonomic innervation of subcutaneous and visceral fat[26,27]. Both fat depots are innervated by distinct sympathetic and parasympathetic motor neurons with functional consequences of autonomic balance on adipocyte insulin sensitivity and energy storage[28]. The proximal regulatory neurons that project into the adipose tissue appear to reside in the hypothalamus[27,29]. Thus, changes in autonomic nervous system

balance that are induced by brain insulin action[8,13,14,24,25] could exert differentially effects on subcutaneous and visceral adipocytes and thereby contribute to our current findings. In line with this hypothesis, induction of brain insulin action was found to modulate systemic but not subcutaneous lipolysis in humans[10].

Our current findings are of particular importance, given that the enlarged visceral fat content not only poses a high risk factor for the subsequent development of diabetes, but is also robustly linked to the risk of cardiovascular disease and the development of cancer[30]. Brain insulin resistance therefore seems to be involved in the pathogenesis of obesity in general. More importantly, it appears to be a determinant of healthy and unhealthy obesity.

Unfortunately, only a limited sample size was available in our longitudinal cohort. Furthermore, we used two different measurement techniques to assess the brain's response to insulin. MEG and fMRI most likely capture different aspects of brain insulin sensitivity and their comparability in this regard has not been tested so far.

In conclusion, we showed that high brain insulin sensitivity was linked to weight loss during lifestyle intervention and associates with a favorable body fat distribution. Our current results underline the importance of brain insulin action for the development of body weight and body fat distribution. As visceral

fat is strongly linked to diabetes, cardiovascular risk, and cancer, these findings have implications beyond metabolic diseases and indicate the necessity of strategies to resolve insulin resistance of the human brain.

## Methods

**Participants and study design of the longitudinal study.** Details on the TULIP lifestyle intervention study, including primary and secondary outcomes as well as inclusion and exclusion criteria, have been reported previously[31]. The program which combined increased physical activity with low fat/high fiber diet started with a 9 month intensive phase and went on for 24 months. The study was conducted within the Deutsche Forschungsgemeinschaft (DFG) project KFO 114. Three hundred participants at high risk for type 2 diabetes completed the intervention. As reported in ref. [31], a group of 190 individuals was re-examined around 9 years later.

In a subgroup of 28 participants, brain insulin sensitivity was assessed by MEG before lifestyle intervention. Of these, 15 individuals were followed-up after $9.6 \pm 0.8$ years (mean ± SEM; for patient characteristics, see Supplementary Table 1). Total dietary energy intake was assessed in 10 of these participants at three time periods during the lifestyle intervention (before, during the first 9 months of lifestyle intervention, and during month 9–24 of lifestyle intervention) by the mean values of several 3-day food diaries obtained at each visit[31].

**Participants of the cross-sectional study.** In a cross-sectional study, in 112 participant's brain insulin sensitivity was assessed by fMRI with administration of 160 U insulin as nasal spray[32]. In all participants, body fat distribution was assessed by whole-body MRI as part of the baseline examination of clinical trials (clinicaltrials.gov: NCT03227484, NCT02870361, NCT02991365, NCT01797601, NCT01847456, NCT02468999). For patients characteristics, see Supplementary Table 2.

**Determination of body fat distribution by MRI.** Whole-body MRI was performed in the early morning after overnight fasting on a 1.5 T whole-body imager (Magnetom Sonata, Siemens Healthcare, Erlangen, Germany). A T1-weighted fast-spin-echo technique (echo-time TE = 12 ms, repetition time TR = 490 ms) was applied, acquiring axial images with a slice-thickness of 10 mm and an interslice gap of 10 mm. Depending on the size of the subject, 100–120 images were recorded from toes to fingers with subjects being in prone position with extended arms in a total measuring time of 20–25 min. Detailed information is given in ref. [33] Segmentation and quantification of adipose tissue compartments was performed by an automated fuzzy-clustering algorithm with orthonormal snakes[34]. Visceral adipose tissue was in the abdominal cavity between femoral heads and thoracic diaphragm, subcutaneous adipose tissue along the body axis from feet to pelvis (available in $N = 99$).

**Brain insulin sensitivity in the longitudinal study.** Before lifestyle intervention, participants underwent two hyperinsulinaemic–euglycaemic glucose clamps with insulin or placebo (saline) infusion on two different days (for details see Tschritter et al.[20,35]). Cerebrocortical activity was assessed by magnetoencephalography (MEG) before and during the clamp experiment. The power spectrum for the spontaneous activity of the participants was analyzed by a standard statistical mapping procedure taking into account multiple comparison correction for the different frequency bands. On the basis of earlier findings[20], assessment of the cerebrocortical insulin effect as changes in theta activity during the insulin experiment corrected for the placebo experiment were calculated[35].

**Hypothalamic insulin sensitivity in the cross-sectional study.** Participants underwent whole-brain fMRI at a 3.0 T scanner (Siemens MAGNETOM Prisma, Erlangen, Germany) to assess regional insulin sensitivity of the brain, as recently reported[22]. Experiments were conducted after an overnight fast and started under basal condition to quantify cerebral blood flow (CBF) with a pulsed arterial spin labeling (PASL) measurement using a PICORE-Q2TIPS sequence (proximal inversion with control for off-resonance effects—quantitative imaging of perfusion using a single subtraction). Following the basal measurement, 160 U of human insulin were administered as nasal spray[13]. After 30 min, PASL was assessed a second time. Baseline-corrected CBF maps were computed to quantify the CBF changes 30 min after intranasal insulin administration. Change in CBF was extracted from the hypothalamus based on recent findings[22].

**Informed consent.** All relevant ethical regulations were complied with and informed written consent was obtained from all participants. The local ethics committee approved the study protocols (Ethics Committee of the Medical Faculty of the Eberhard-Karls-Universität and the University Hopsital Tübingen).

**Statistical analyses.** Unless otherwise stated, data are given as mean ± SEM. The software package used was JMP 13 (SAS Institute, Cary, NC, USA) and a $p$ value ≤ 0.05 was considered statistically significant.

In the longitudinal study, changes in body weight, body fat depots, and further metabolic variables and their association with baseline brain insulin sensitivity (theta activity) were analyzed by MANOVA. Continuous variables were used for analyses and stratified variables were used solely for better illustration of the results.

For the cross-sectional study, correlations between body fat compartments and hypothalamic cerebral blood flow (fMRI measurements) were analyzed by linear regression models unadjusted and adjusted for sex and age as well as BMI.

**Reporting summary.** Further information on research design is available in the Nature Research Reporting Summary linked to this article.

## Data availability

The data that support the findings of this study are available on reasonable request from the corresponding author M.H. The data are not publicly available due to them containing information that could compromise research participant privacy/consent. The source data underlying Figs. 1a–c and 2b–d are provided as a Source Data file.

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

## Acknowledgements
We thank A. Bury, M. Borutta, E. Kollmar, and A. Vosseler for their excellent technical assistance. The study was partly supported by grants from the Deutsche Forschungsgemeinschaft to A.F. and by the German Federal Ministry of Education and Research (BMBF) to the German Centre for Diabetes Research (DZD, 01GI0925).

## Author contributions
S.K., R.W., and V.V. researched and analyzed data; O.T. and J.M. researched data. H.U.H., H.P., and A.F. supervised the project. M.H. researched data, supervised the project, and drafted the manuscript together with S.K. and V.V. All authors contributed to the discussion and approved the final version of the manuscript prior to submission.

## Competing interests
The authors declare no competing interests.
