## [Peer Review File · Nature Communications]

Reviewers' comments:

Reviewer #1 (Remarks to the Author):

The manuscript by Kullmann S et al. reports data on two cohorts of individuals, the first longitudinal on 15 subjects with brain insulin sensitivity determined by magnetoencephalography before a lifestyle intervention program, and the second cross-sectional on 112 participants with functional MRI of the hypothalamus. The first cohort was already assessed at earlier time points in a previous report from the same group.

The Authors show that subjects with high brain insulin sensitivity prior to lifestyle intervention lost more body weight and during the 24 months of the program. In addition, high brain insulin sensitivity was associated with lower body weight regain and body fat in the long-term follow-up.

Specific issues.

1. Indeed, while subjects are defined as “insulin sensitive” or “insulin resistant”, there is no information about which thresholds were used for these definitions in relation to insulin-mediated changes in theta activity determined by magnetoencephalography and changes in hypothalamic blood flow determined by functional MRI.

2. It is unclear why the Authors state that “especially visceral fat was lost during the 24 months intervention” (line 107) and that “this association was present for the amount of visceral, but not with subcutaneous fat in our longitudinal data” (lines 146-147). Indeed, in the longitudinal study greater loss of both visceral and subcutaneous fat was found in subjects with higher insulin sensitivity, and indeed this was quantitatively more for the subcutaneous fat (Suppl. Table 3). In general, why there is loss of both subcutaneous and visceral fat in the longitudinal study but association of only visceral fat in the cross-sectional study is unclear. It is true that in one study brain insulin action was defined as changes in theta activity determined by magnetoencephalography following an insulin clamp, while in the other study as changes in hypothalamic blood flow determined by functional MRI following nasal administration of insulin. However, how do these two measures of insulin action differ qualitatively and quantitatively is not known. Would changes in hypothalamic blood flow predict any extent of weight loss following a lifestyle intervention?

3. The results of this study are of interest. However, it remains unknown how whole brain insulin action may impact on weight loss/regain following lifestyle intervention and similarly which mechanism may explain the association between hypothalamic responses to insulin and the amount of visceral fat. The Authors do not provide any information on (nor discuss) important factors such as compliance to diet and actual dietary intake, level of physical activity, and energy expenditure.

4. Another important unaddressed question is the impact on excess visceral fat on metabolic/inflammatory markers, which could have been easily measured.

5. The 15 individuals examined in the longitudinal study represent a very small cohort compared to the entire group exposed to the lifestyle intervention. Did these individuals differ in anthropometric, biochemical or behavioral characteristics from those in the larger cohort?

6. Assessment of brain insulin sensitivity was carried out only once at baseline. Given the long follow-up, it is possible that insulin sensitivity may have changed over time. Was this assessed?

7. Lines 145-146: it should read: "low brain insulin sensitivity predicts a regain in body weight and body fat in the 9 years following the program".

Reviewer #2 (Remarks to the Author):

This interesting paper describes two different studies investigating the effects of brain insulin sensitivity on weight loss during a weight loss intervention and the possible associations with visceral and subcutaneous fat.

The first study, which looked at weight loss during a lifestyle intervention, used cerebral magnetoencephalography to assess brain insulin sensitivity and the data suggest that those with greatest brain insulin sensitivity lost more weight and body fat during the study, and also gained less weight during follow up, although numbers were small.

The second, larger, cross-sectional study measured brain insulin sensitivity using functional MRI responses to intranasal insulin and related this to visceral and subcutaneous fat measured using MRI; this suggested that higher brain insulin sensitivity was associated with low amounts of visceral fat.

Given that obesity, and particularly visceral fat, is already known to be associated with whole body insulin sensitivity it is perhaps not surprising to find this association. The paper would be much stronger if some measure of whole body insulin sensitivity was included (ideally clamp-derived, but surrogate measures such as HOMA-IR or Matsuda index would also be of interest). These data should be included if available – the authors mention that clamps were done as part of the weight loss study so the clamp-derived measures of whole body insulin resistance should certainly be reported or referenced.

Whilst the association of brain insulin resistance with visceral fat is perhaps unsurprising, the fact that it seems to predict response to a lifestyle intervention is of interest. Could the authors speculate about possible mechanisms here, and perhaps consider and discuss whether there is other evidence that those who are insulin resistant find it more difficult to lose weight, as there is no clear consensus that is the case.

John Wilding

Reviewer #3 (Remarks to the Author):

Review Nature Communications NCOMMS-19-18496-T

1. I feel “predict’ may be too strong of a word for a the type of research design completed. There is strong evidence for an association but as in most human studies there are not enough experimental control to make causal statements.

2. Table 3 should report sample sizes in the header (top row columns 2 & 3) of the table for each group (ie, brain insulin resistant; brain insulin sensitive).

3. In terms of the statistical analyses for the data reported in Table 3, some reviewers might note that the MANOVA approach to analyzing repeated measures data is not the most powerful approach. I contend in this case, that most of the effects were significant, despite using a procedure with lesser power. Furthermore, give the time gaps in the data collection (ie, 15 between 9 and 24 months, 7 years between 24 months and 9 years) the covariances are probably unstructured and the MANOVA approach is appropriate.

4. The authors should report the sample size (N) and correlation coefficient (r) for the scatterplots in Figure 2. The values should appear in either the text of the manuscript, Figure Legend, and/or on the graphs themselves.

5. Since the results for the correlations mention that the VAT-Blood Flow relationship was significant after adjustment for BMI (line 131), BMI should be mentioned as a control covariate in the Methods section (line 238).

6. The results for the correlations mention that the VAT-Blood Flow relationship was significant after adjustment for BMI (line 131), and significant after adjustment for sex and age (line 132). Was it significant after adjustment for all 3 of these covariates?

7. Similarly, the results for the correlations mention that the VAT/SCAT-Blood Flow relationship was significant (line 136). Was it significant after adjustment for BMI, sex, and/or age as covariates?

We thank you for giving us the opportunity to revise and improve our manuscript. We thank the reviewers for the encouraging comments. The suggestions substantially improved the manuscript. Please find below a point-to-point response. Changes in in the manuscript are marked by using red fonts.

Reviewer #1 (Remarks to the Author):

The manuscript by Kullmann S et al. reports data on two cohorts of individuals, the first longitudinal on 15 subjects with brain insulin sensitivity determined by magnetoencephalography before a lifestyle intervention program, and the second cross-sectional on 112 participants with functional MRI of the hypothalamus. The first cohort was already assessed at earlier time points in a previous report from the same group.

The Authors show that subjects with high brain insulin sensitivity prior to lifestyle intervention lost more body weight and during the 24 months of the program. In addition, high brain insulin sensitivity was associated with lower body weight regain and body fat in the long-term follow-up.

Specific issues.

1. Indeed, while subjects are defined as “insulin sensitive” or “insulin resistant”, there is no information about which thresholds were used for these definitions in relation to insulin-mediated changes in theta activity determined by magnetoencephalography and changes in hypothalamic blood flow determined by functional MRI.

Thank you for this comment, we acknowledge that this was not stated clear enough in our initial version of the manuscript. We apologize for this. The statistical analyses were performed using a continuous variable for brain insulin resistance, thus, for statistical analyses participants were not divided in “insulin sensitive” and “insulin resistant”. To visualize the results of the MONAVO in the longitudinal cohort, we split the group by the median of insulin-induced change in the theta frequency band (MEG data). This is now clearly reported in the Methods section as well as in the figure legend.

2. It is unclear why the Authors state that “especially visceral fat was lost during the 24 months intervention” (line 107) and that “this association was present for the amount of visceral, but not with subcutaneous fat in our longitudinal data” (lines 146-147). Indeed, in the longitudinal study greater loss of both visceral and subcutaneous fat was found in subjects with higher insulin sensitivity, and indeed this was quantitatively more for the subcutaneous fat (Suppl. Table 3). In general, why there is loss of both subcutaneous and visceral fat in the longitudinal study but association of only visceral fat in the cross-sectional study is unclear. It is true that in one study brain insulin action was defined as changes in theta activity determined by magnetoencephalography following an insulin clamp, while in the other study as changes in hypothalamic blood flow determined by functional MRI following nasal administration of insulin. However, how do these two measures of insulin action differ qualitatively and quantitatively is not known. Would changes in hypothalamic blood flow predict any extent of weight loss following a lifestyle intervention?

We apologize for the non-precise wording. We removed one of the respective sentences and rephrased the other. The reviewer is right that more subcutaneous than visceral fat was lost during the lifestyle intervention. However, brain insulin responsiveness was associated with the loss of total and visceral fat ($p=0.01$ and $p=0.009$, sup. table 3), this did not reach statistical significance for subcutaneous fat, neither unadjusted ($p=0.07$), nor after adjustment for the loss in total adipose tissue ($p=0.2$). The association with loss in visceral fat also remained significant after adjustment for change in total adipose tissue ($p=0.007$). This fits well to the results of our cross-sectional cohort where we detected associations only with visceral but not subcutaneous fat.

The reviewer addresses the important point of the two different measurement approaches for brain insulin responsiveness. Magnetoencephalography measures directly neuronal activity generated in synchronously active ensembles of neurons. In the original study we reported that insulin induced theta band activity correlated with loss in visceral fat, however we were not able to determine the exact location of the generating source. Due to this limitation and the

limited resolution of MEG for deeper brain structures, i.e. hypothalamus etc., we implemented fMRI to detect insulin action in the brain in future studies. In general, it is not possible to measure MEG and fMRI simultaneously. Based on this we are not able to provide further insights in the link between directly measured neuronal activity by MEG and the proxy measurement of brain activity by CBF. Arterial spin labelling was used for functional MRI, which is a quantitative measure of cerebral blood flow (CBF). The exact mechanisms of the neurovascular coupling, i.e. the relation of cortical MEG and CBF, is still an unresolved issue. It can be assumed that insulin affects neuronal activity and CBF through different pathways. In MEG we assume that insulin affects neuronal synaptic plasticity and activity, which results in changes in CBF. It is also possible that insulin additionally affects activity in astrocytes, which contribute to the cBF signal. In this respect, we are not able to make any conclusive statements about the link between theta band activity measured with MEG and CBF. The question of the reviewer is definitely an important question in the field for the future and we now clearly addressed this point in the limitations paragraph of the Discussion.

3. The results of this study are of interest. However, it remains unknown how whole brain insulin action may impact on weight loss/regain following lifestyle intervention and similarly which mechanism may explain the association between hypothalamic responses to insulin and the amount of visceral fat. The Authors do not provide any information on (nor discuss) important factors such as compliance to diet and actual dietary intake, level of physical activity, and energy expenditure.

As suggested, we now included a new paragraph on possible underlying mechanisms to the Discussion section of the manuscript. We agree with the reviewer that there is accumulating evidence that brain insulin action affects dietary intake, possible physical activity and energy expenditure. However, as we assume that these effects of brain insulin will likely impact on body weight but not directly influence body fat distribution, we focused on the role of the autonomic nervous system in our new paragraph as autonomic nervous differentially innervate subcutaneous and visceral fat and could therefore possibly explain the fat depot specific associations that we detected.

As suggested by the reviewer, we analyzed data on dietary intake and physical activity in the longitudinal study. These data were unfortunately only available during the 24 months of the lifestyle intervention. Interestingly, we detected an association between insulin effects in the brain and subsequent changes in total dietary intake (assessed by food diaries) with a stronger reduction in dietary intake in the brain insulin sensitive participants. This is well in line with the known anorexigenic action of insulin in the brain. We now included this information in the results section of the manuscript.

For physical activity (assessed as habitual physical activity (HPA) score), no such association was present.

4. Another important unaddressed question is the impact on excess visceral fat on metabolic/inflammatory markers, which could have been easily measured.

Thank you for this comment. Unfortunately, neither associations between brain insulin sensitivity and changes in glycemia nor with hs-CRP reached statistical significance in the longitudinal cohort (all $p \geq 0.178$). Of course, this might be due to the limited statistical power due to the smaller sample size of the longitudinal cohort.

Though, in the larger cross-sectional cohort, we detected associations between hypothalamic insulin responsiveness and glycemia and whole-body insulin sensitivity. This is now included in the results part of the manuscript.

5. The 15 individuals examined in the longitudinal study represent a very small cohort compared to the entire group exposed to the lifestyle intervention. Did these individuals differ in anthropometric, biochemical or behavioral characteristics from those in the larger cohort?

We absolutely agree with the reviewer that the sample size in the longitudinal cohort is a limitation of our study. This is now specifically addressed in the limitations part of the manuscript.

The participants who underwent MEG measurement of brain insulin sensitivity were comparable to the entire group in age ($p=0.07$), BMI ($p=0.7$), fasting glycemia ($p=0.2$), glucose

tolerance (p=0.2), fasting insulin (p=0.7), OGTT-derived insulin sensitivity index (p=0.9), amount of total (p=0.4), visceral (p=0.3), and subcutaneous (p=0.3) adipose tissue, but had a higher waist-to-hip ratio (0.89 vs 0.95; p=0.04) and a lower HbA1c (5,67 % vs 5,45 % ; p=0.03).

We have not yet included this information in the manuscript. However, if the reviewer feels that this information will help readers in the interpretation of our results, we are of course ready to put it in.

6. Assessment of brain insulin sensitivity was carried out only once at baseline. Given the long follow-up, it is possible that insulin sensitivity may have changed over time. Was this assessed?

Indeed, we tried to assess this over time. Unfortunately, the number of participants in whom this measurement could be obtained was even lower than the number of participants reported in the current manuscript. Due to the insufficient number of available measurements, we decided to not follow this further.

7. Lines 145-146: it should read: "low brain insulin sensitivity predicts a regain in body weight and body fat in the 9 years following the program".

Thanks, we corrected the sentence.

Reviewer #2 (Remarks to the Author):

This interesting paper describes two different studies investigating the effects of brain insulin sensitivity on weight loss during a weight loss intervention and the possible associations with visceral and subcutaneous fat.

The first study, which looked at weight loss during a lifestyle intervention, used cerebral magnetoencephalography to assess brain insulin sensitivity and the data suggest that those with greatest brain insulin sensitivity lost more weight and body fat during the study, and also gained less weight during follow up, although numbers were small.

The second, larger, cross-sectional study measured brain insulin sensitivity using functional MRI responses to intranasal insulin and related this to visceral and subcutaneous fat measured using MRI; this suggested that higher brain insulin sensitivity was associated with low amounts of visceral fat.

Given that obesity, and particularly visceral fat, is already known to be associated with whole body insulin sensitivity it is perhaps not surprising to find this association. The paper would be much stronger if some measure of whole body insulin sensitivity was included (ideally clamp-derived, but surrogate measures such as HOMA-IR or Matsuda index would also be of interest). These data should be included if available – the authors mention that clamps were done as part of the weight loss study so the clamp-derived measures of whole body insulin resistance should certainly be reported or referenced.

Thank you for this comment. In the group of participants of whom MEG assessment of brain insulin sensitivity was available, neither OGTT-derived nor clamp-derived insulin sensitivity index was linked to the loss of total fat mass or visceral fat during the lifestyle intervention (all $p \geq 0.4$).

However, when we analyzed this relation in the entire group of participants in the lifestyle intervention program and available long-term data (N=190), we detected associations with OGTT-derived insulin sensitivity (Matsuda): Good peripheral insulin sensitivity was associated with a smaller loss of body weight and body fat during the first 9 months of the lifestyle intervention. However, for the subsequent part of the lifestyle intervention and the long-term follow up, no association with baseline Matsuda index was present.

As our manuscript is mainly focused on brain insulin sensitivity and in order to keep the manuscript easy to follow, we did not yet include this information to the text. However, if the reviewer feels that this information could be useful for the interpretation of our results, we are of course ready to put it in.

Whilst the association of brain insulin resistance with visceral fat is perhaps unsurprising, the fact that it seems to predict response to a lifestyle intervention is of interest. Could the authors speculate about possible mechanisms here, and perhaps consider and discuss whether there is other evidence that those who are insulin resistant find it more difficult to lose weight, as there is no clear consensus that is the case.

The reviewer addresses an important point. We now analyzed data on food intake from our lifestyle intervention. We found that brain insulin responsiveness was linked to a stronger reduction of energy intake during the lifestyle intervention. While the effects of brain insulin action on food intake are established in the literature, they can most likely not explain the preferential loss of visceral fat. As suggested by the reviewer, we now included an entire new paragraph on possible mechanisms to the Discussion section of the manuscript.

Reviewer #3 (Remarks to the Author):

Review Nature Communications NCOMMS-19-18496-T

1. I feel “predict” may be too strong of a word for a the type of research design completed. There is strong evidence for an association but as in most human studies there are not enough experimental control to make causal statements.

We agree with the reviewer that our current study cannot ultimately establish causality. As suggested, we tempered the wording accordingly and removed “predict” from the text.

2. Table 3 should report sample sizes in the header (top row columns 2 & 3) of the table for each group (ie, brain insulin resistant; brain insulin sensitive).

This was added to the table.

3. In terms of the statistical analyses for the data reported in Table 3, some reviewers might note that the MANOVA approach to analyzing repeated measures data is not the most powerful approach. I contend in this case, that most of the effects were significant, despite using a procedure with lesser power. Furthermore, give the time gaps in the data collection (ie, 15 between 9 and 24 months, 7 years between 24 months and 9 years) the covariances are probably unstructured and the MANOVA approach is appropriate.

Thank you for the thorough overlook on our statistical methods.

4. The authors should report the sample size (N) and correlation coefficient (r) for the scatterplots in Figure 2. The values should appear in either the text of the manuscript, Figure Legend, and/or on the graphs themselves.

Thanks for pointing out this missing information. We now included the sample size to the figure legend and the correlations coefficients (r) to the figure. Furthermore, we now report r^2 in the Results section of the manuscript.

5. Since the results for the correlations mention that the VAT-Blood Flow relationship was significant after adjustment for BMI (line 131), BMI should be mentioned as a control covariate in the Methods section (line 238).

This is now mentioned in the Methods section.

6. The results for the correlations mention that the VAT-Blood Flow relationship was significant after adjustment for BMI (line 131), and significant after adjustment for sex and age (line 132). Was it significant after adjustment for all 3 of these covariates?

After adjustment for all three covariates, this no longer reached statistical significance ($p=0.1$)

7. Similarly, the results for the correlations mention that the VAT/SCAT-Blood Flow relationship was significant (line 136). Was it significant after adjustment for BMI, sex, and/or age as covariates?

This remained statistically significant after adjustment for BMI as well as sex and age (now reported in the manuscript) but not after adjustment for all three.

REVIEWERS' COMMENTS:

Reviewer #1 (Remarks to the Author):

No further comment.

Francesco Giorgino

Reviewer #2 (Remarks to the Author):

Thank you for considering and responding to the suggestions for revision. These have been adequately addressed.

Reviewer #3 (Remarks to the Author):

I think the authors did a good job responding to my suggestions. And from my reading, I think they addressed the other reviewers concerns.

We are happy that the reviewers were satisfied with the changes we made in the first revision round.

Therefore, no point-to-point to the reviewers was necessary.